# Experimental evaluation of a metofluthrin passive emanator against *Aedes albopictus*

**Olivia Zarella**[1], **Uche Ekwomadu**[1], **Yamila Romer**[1], **Oscar D. Kirstein**[1], **Azael Che-Mendoza**[2], **Gabriela González-Olvera**[2], **Pablo Manrique-Saide**[2], **Gregor Devine**[3], **Gonzalo M. Vazquez-Prokopec**[1]*

1 Department of Environmental Sciences, Emory University, Atlanta, GA, United States of America, 2 Unidad Colaborativa de Bioensayos Entomológicos, Campus de Ciencias, Biológicas y Agropecuarias, Universidad Autónoma de Yucatán, Mérida, Yucatán, México, 3 Mosquito Control Laboratory, QIMR Berghofer Medical Research Institute, Brisbane, Queensland, Australia

* gmvazqu@emory.edu

## Abstract

### Background

With the recent occurrence of locally transmitted *Aedes*-borne viruses in the continental United States and Europe, and a lack of effective vaccines, new approaches to control *Aedes aegypti* and *Aedes albopictus* are needed. In sub-tropical urban settings in the US, *Ae. albopictus* is a dominant nuisance and arbovirus vector species. Unfortunately, the vector control toolbox against *Ae. albopictus* is not as well developed as for *Ae. aegypti*. Here, we evaluate the efficacy, longevity, and range of protectiveness of a novel passive metofluthrin emanator (10% active ingredient in a polyethylene mesh) against *Ae. albopictus* indoors and outdoors.

### Methods

Four studies were conducted comparing the presence of the metofluthrin emanator to a control lacking emanator with interest in quantifying efficacy by human landing counts. Studies evaluated the effect of an emanator at varying distances from one or more human volunteers indoors and outdoors. Efficacy of emanators over time since activation was also evaluated.

### Results

Mixed-effects models determined that sitting in close proximity to an emanator reduced landings by 89.5% outdoors and by 74.6% indoors. The emanator was determined protective when located immediately next to a human volunteer outdoors but not uniformly protective when located further away. The emanator was protective at all tested distances from the device indoors. Mortality of mosquitoes exposed to metofluthrin emanators was ~2x higher than those who were not exposed in indoor conditions. Finally, a Generalized Additive Model determined that emanators used continuously outdoors lost their effect after 2.5 weeks and stopped inducing paralysis in mosquitoes after 3.8 weeks of use.

**Data Availability Statement:** Emanators supplied by Sumitomo Chemical Company Ltd. Data supporting the conclusions of this article are included within the article. Data generated or

analyzed during the present study are publicly available from: Vazquez-Prokopec, Gonzalo (2021), "Experimental evaluation of a metofluthrin passive emanator against Aedes albopictus", Mendeley Data, V1, doi: 10.17632/dkdc8642cv.1.

**Funding:** This study was supported by a grant from USAID (AID-OAA-F-16-00094; Devine, PI). The funders had no role in study design, data collection and analysis, decision to publish, or preparation of the manuscript.

**Competing interests:** The authors declare that they have no competing interests.

## Conclusions

We show strong and lasting efficacy of 10% metofluthrin emanators against field *Ae. albopictus* both in indoor and outdoor conditions. Metofluthrin emanators can protect people from *Ae. albopictus* bites, representing a viable option for reducing human-mosquito contacts at home and beyond.

## Introduction

The rapid expansion and dramatic global increase in the burden of dengue, Zika and Chikungunya viruses constitutes a major public health problem in tropical and subtropical urban areas [1–4]. Transmitted primarily by *Ae. aegypti* and, to a lesser extent, *Ae. albopictus*, control of these viruses depends primarily on health education and vector control, as no effective vaccine or prophylactic measure is yet available [1, 5]. While most of Europe and the continental US have been free of local *Aedes*-borne virus transmission for decades, the last twenty years have seen a rapid increase in locally acquired cases. Traditionally affecting U.S. territories, such as Hawaii, the Virgin Islands, and Puerto Rico [6–8], dengue outbreaks have very frequently occurred in Florida [9] and Texas [10, 11]. Later, outbreaks of chikungunya [12] and Zika [13] in Florida, pointed out the potential role of *Ae. albopictus* in enhancing local transmission [14, 15] in areas where both *Ae. aegypti* and *Ae. albopictus* overlap [16]. Recent outbreaks in Europe and other parts of the world have incriminated *Ae. albopictus* as the main vector [17]. *Ae. albopictus* was associated with significant outbreaks, including the 2005–2006 chikungunya outbreak on the island of Réunion [18] or the 2017 chikungunya outbreak in the Lazio and Calabria regions of Italy [19]. With an increasing number of epidemics linked to *Ae. albopictus*, future control efforts should aim to target more aggressively this vector in areas prone to local arbovirus transmission.

Unfortunately, the knowledge base supporting the effectiveness of available control tools for *Ae. albopictus* control is less developed than for *Ae. aegypti*. Differences among species in spatial distribution, habitat use within cities, and adult female densities indoors and outdoors, human biting rate [20–24] may limit the effectiveness of methods known to impact *Ae. aegypti* on *Ae. albopictus*. For instance, while outdoor barrier spraying may not be effective against *Ae. aegypti*, it is the mainstay of *Ae. albopictus* control in the Torres Strait, Australia, due to its outdoor resting behaviour [25].

Volatile pyrethroids such as transfluthrin and metofluthrin have potential for the control of *Ae. aegypti* when incorporated into passive emanators (insecticide-treated materials that slowly release the active ingredient [a.i.] into the air, without the need for power or heat). These chemicals are captured under the "spatial repellent" paradigm and exert a variety of dose-dependent lethal, repellent and confusant effects on a variety of insects [26, 27]. Under laboratory conditions, metofluthrin is effective (in toxicity and biting inhibition) to *Ae. albopictus* [28]. When used as a mosquito coil and compared to other commonly used volatile pyrethroid a.i. such as transfluthrin and d-allethrin under laboratory conditions, metofluthrin had greater chemical stability [29] and produced the highest *Ae. albopictus* mortality rate (> 80%) [30]. A manufactured passive emanator consisting of a fan-type paper device with a non-heated metofluthrin formula was highly effective against *Ae. albopictus*, reducing biting by 92–97%, but with its effectiveness rapidly decreasing after 48 hours due to the rapid loss of a.i. from the paper matrix [31]. An engineered alternative to this passive emanator, consisting of a resin formulation containing 4.4% metofluthrin, has proven more durable and effective at volatilizing

the a.i. compared to paper or other carriers [29]. More recently, a prototype passive emanator containing 10% metofluthrin w/w (SumiOne®, Sumitomo Chemical) has been extensively tested against *Ae. aegypti*, showing significant impact on indoor abundance and biting [26]. The device consists of an impregnated net housed in a rectangular plastic cartridge designed to be hung indoors using an integral hook. Under semi-field conditions, exposure to the emanator for one hour reduced landing behavior for pyrethroid-susceptible *Ae. aegypti* mosquitoes from 32–46 landings to zero [32]. Also, 80–90% of mosquitoes were knocked down during that time. An entomological randomized field trial conducted in Ticul, Mexico, found that placing emanators at a rate of one per room per house led to a reduction in indoor female *Ae. aegypti* density of 70% and of >90% reduction in landings compared to houses without emanators [33].

In contrast to the increasing volume of work on the effectiveness of volatile pyrethroids against *Ae aegypti* indoors, there remains a lack of empirical study on their impact indoors or outdoors on *Ae. albopictus* mosquitoes. This study evaluated the efficacy of a 10% w/w/ metofluthrin emanator against pyrethroid susceptible *Ae. albopictus* populations from Southeastern U.S. Specifically, we quantified the impact of emanators on landing behavior, mortality and repellency of *Ae. albopictus* upon exposure to the device in laboratory, semi-natural, and field conditions.

## Methods

### Study design

The study involved four independent evaluations that occurred either indoors or outdoors depending to the specific hypothesis to be tested. Indoors testing occurred inside a bedroom from a house located near Emory University. Field testing was conducted in Baker Woodlands, located within Emory University campus in Atlanta, GA (33˚47'20.5 "N, 84˚19'34.3 "W), a landscape densely populated with *Ae. albopictus*. The human landing count (HLC) method is the gold standard for determining human-mosquito contact rates [34] and was used to quantify the efficacy of the emanators. For our study, a volunteer sat in a camping chair wearing a bug-proof jacket, hood and long socks. The skin on their legs from knees to ankles or arm from hand to elbow (depending on the evaluation) was exposed. During data collection, the technician prevented mosquito biting by brushing away mosquitoes immediately after they landed (33). Each landing was counted for a 5-minute period. For experiments in tents (84" x 84" x 48") and bedroom, an assessment of Knock-down was also made by counting mosquitoes exhibiting unusual behavior of being "knocked out" of the air due to paralysis from exposure to the insecticide; representing the mosquito's inability to stand, fly, or take-off [35, 36].

**Study 1: Testing the short-distance effect of emanators on *Ae. albopictus* landing behavior outdoors.** To test whether close proximity to an emanator impacts *Ae. albopictus* landing behavior outdoors, a field-trial was designed and applied on four separate days from June 2017 to July 2017. The field-trial consisted of four HLC collections per day (2 metofluthrin treatment and 2 control). A total of 8 tests were conducted per arm, between 8AM to 11AM, and spaced 1 h apart (Fig 1A). Test treatments were randomized across the 5 min assessments and trials occurred during the summer on days with no rain. Before each trial, the technician sat with unexposed skin for 5 min to acclimate and attract mosquitoes in the area.

**Study 2. Testing emanator range of activity on *Aedes albopictus* outdoors.** Study 1 provided the efficacy of emanators at immediate distance (<1m). To determine the distance over which the emanators were effective outdoors, we examined *Ae. albopictus* landing behavior within a 3-8m radius of the emanator. The field-trial was conducted in the morning, over nine days from August to September 2018. Two locations in Baker Woodlands with similar

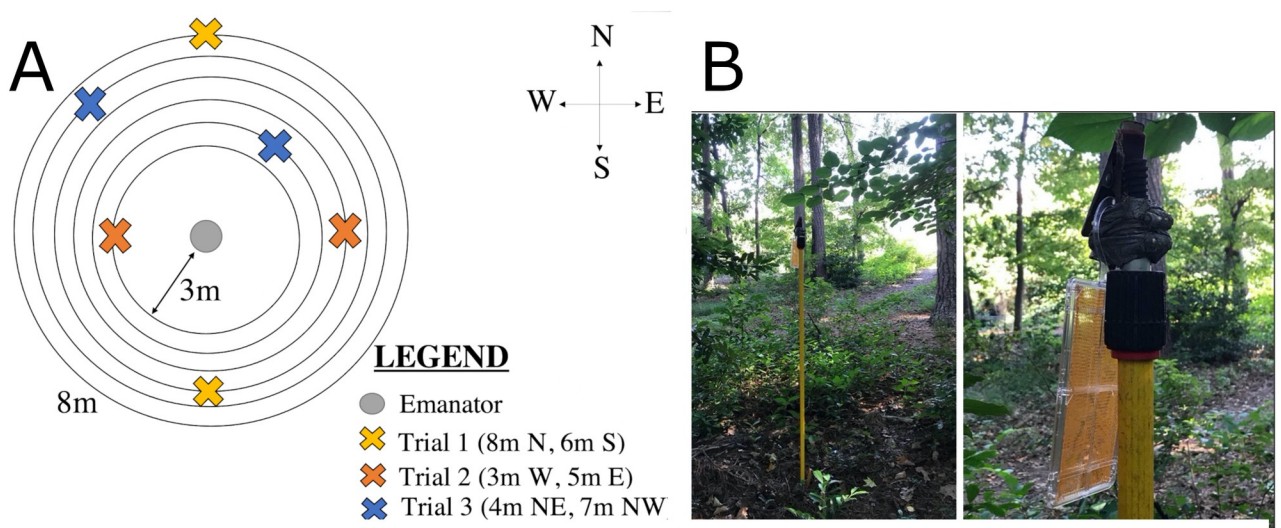

**Fig 1. Diagram of different studies conducted to evaluate the efficacy of metofluthrin emanators on *Ae. Albopictus*.** (A) Metofluthrin emanators were hung outdoors on top of a 1 m stick (A) and used to test their effect on human landing counts in the immediate vicinity (<1m distance). (B) Emanators were hung at increasing distances from it up to 8 meters. (C) An outdoor tent experiment was set outdoors to evaluate the impact of emanators on mosquito entry and landing counts. (D) An indoor evaluation of the impact of emanators on *Ae. albopictus* landing counts and knock down was conducted inside a bedroom and placing mosquitoes in bugdorms at 1m and 3m from the emanator.

vegetation and *Ae. albopictus* characteristics, 20m apart, were selected. Each day, a trial consisting of a matched control (no emanator) and treatment (emanator) was conducted in a pair of distances from 3 to 8 meters selected at random. We tested efficacy at 3, 4, 5, 6, 7 and 8 m from the emanator. The same distances were tested in each location, and the assignment of treatment to each location was also selected at random. Four replicates for each distance in each arm (treatment and control) were conducted. Two collectors participated in this study, randomized between treatments and locations. HLC quantification occurred as above. Factors for analysis considered for each test included cardinal direction, distance, time of day of each test, collection period, HLC, identity of the observer, temperature, humidity, and wind speed and direction.

**Study 3: Testing the effect of metofluthrin emanators on entry behavior.** To test whether metofluthrin emanators impact *Ae. albopictus* entry and landing behavior inside tents, a trial was conducted outdoors during the morning over four days from August to September 2018. Two four-person tents (Coleman Sundome, Coleman Company Inc.) were set ~20 m apart, with a seated observer inside to assess HLC and a white tarp placed on the floor to assess knock-down (Fig 1C). Tent treatments and locations were randomized between tests and two observers were randomly assigned to each tent. Two 10-minute collections were made for each treatment instead of the 5 min HLCs from before to give more time mosquitoes to enter the tents. Before data collection, a metofluthrin emanator was hung from the ceiling of the designated intervention tent, 48" from the ground and above the seated volunteer. The tent entrance was closed for ten minutes with one volunteer inside. After this 10-minute acclimation period, the volunteer opened the tent to allow mosquito host-seeking and entry and conducted a 10-minute HLC collection. Variables collected included time, tent location, temperature, humidity, and wind direction and speed.

**Study 4. Testing emanator longevity and range of activity indoors.** To test how the distance and age of the emanator could affect *Ae. albopictus* landing behavior indoors, we conducted experiments inside a bedroom between October and December 2017. Impacts on landing behavior and knock-down were tested at distances of 1 and 3 meters from the emanator (to capture the extent of the room). We also looked at the effective longevity of the emanators by repeating experiments at weekly intervals and up to 5 weeks.

We followed the design described by Darbro et al. [37] by conducting experiments in a bedroom with dimensions 3.7 m x 2.9 m x 2.4 m. A group of 10 emanators was used throughout the trial and were aged indoors in a ventilated room. Each weekly assessment period, a randomly chosen emanator was used against *Ae. albopictus* female mosquitoes raised from F0 eggs collected in Baker Woodlands. The weekly treatments included four levels: untreated control at 1 m from the emanator, untreated control at 3 m, emanator exposed at 1 m, emanator exposed at 3 m. For each treatment, three bugdorm cages (30 cm x 30 cm x 30 cm) containing ten mosquitoes each were placed at the specified location. A similar number of mosquitoes were kept in the laboratory and was used to determine if transport to the houses in a Styrofoam cooler impacted survival. Before each test, mosquitoes were acclimated to the temperature of the room for 30 minutes.

We ran the evaluations at the time of opening the emanator (t = 0) and then weekly up to week 5. At every time, we quantified mosquito landings on an exposed arm for a total of 2

minutes and knock-down after a 30 min exposure to the emanator. Control trials were conducted before exposure trials to prevent build-up of the active ingredient in the room. To assess mortality, mosquitoes were transferred back to the lab after the completion of each exposure trial and remained in an incubator at 26°C and 80% humidity to assess long-term effects of exposure to metofluthrin. Daily mortality was recorded until 100% of mosquitoes died in both emanator and control treatments.

## Mosquito rearing

Mosquitoes used in the study derived from eggs collected in Baker Woodlands and were reared in the insectary of the Department of Environmental Sciences, Emory University. Briefly, adults and larvae were maintained at $25 \pm 5°C$ and $80 \pm 10\%$ humidity, with a 12:12 (L:D) h photoperiod. Larvae were fed with yeast and liver powder (MP Biomedicals) and adults with cotton soaked in 10% sugar solution. Unmated female adults of 3 to 7 days from emergence were sugar starved for 24 hours before being used in the experiment. Mosquito susceptibility to major insecticide classes was tested by CDC bottle bioassay [38]. Briefly, we coated 25m ml Wheaton bottles with known diagnostic doses of permethrin (15μg/ml), deltamethrin (10 μg/ml) and DDT (75 μg/ml) and quantified mortality/knock-down at the diagnostic time (30 min for deltamethrin and permethrin, 45 min for DDT) and up to 120 min post-exposure. The experiment was replicated 4 times, with each bottle containing 25 recently emerged female *Ae. albopictus*. A control treatment consisted of bottles coated with acetone.

## Statistical analysis

Generalized linear models were used for most analysis and can be viewed in S1 and S2 Tables in S1 File. For HLC, negative binomial generalized linear mixed models (GLMM) were implemented, whereas for studies evaluating emanator longevity a Cox Proportional-Hazards Regression Model (CPHRM) was applied to quantify the probability of mosquito death occurring per day post exposure. Inclusion or exclusion of exploratory variables were determined by using the Akaike Information Criterion (AIC) which compared the model with and without the variable. A summary of the final GLMM model used to assess each hypothesis is found in S1 and S2 Tables in S1 File. For treatments showing a significant reduction in HLC, we estimated the emanator's efficacy, as 1—Incidence Rate Ratios (IRR). This metric, bound between 0 and 1, shows the proportional reduction in a metric compared to the control. All analyses were performed in the R statistical software (version 4.0.3) using the package mgcv.

## Ethics statement

Methods for landing counts were approved by Emory University IRB (IRB00082773: Backyard Sampling of *Aedes* albopictus protocol). Written consent was obtained from all participants

## Results

### *Ae. albopictus* susceptibility to insecticides

At the diagnostic time (30 min for pyrethroids, 45 min for DDT), *Ae. albopictus* from the field (and used in lab experiments) evidenced a mortality of 98.5% for permethrin (diagnostic dose, 15 μg/ml), 100% for deltamethrin (diagnostic dose, 10 μg/ml), and 97% for DDT (diagnostic dose, 75 μg/ml). At 90 minutes post-exposure, all mosquitoes were found dead irrespective of the insecticide. Mortality in the controls was 0% for all insecticides and times.

## Study 1: Testing the short-distance effect of emanators on *Ae. albopictus* landing behavior outdoors

We recorded 932 landing counts over four data collection days with 9.0% (84/932) of landings occurring in the presence of an emanator and 91.0% (846/932) of landings occurring during control periods. The mean landing count of the control was at least 5.3 times the mean landing count of the emanator (Fig 2A), with a statistically significant 89.5% reduction in landing counts in the presence of an emanator, Table 1.

## Study 2. Testing effective distance of emanator activity on *Ae. albopictus* outdoors

We recorded 1,731 landing counts over the course of nine data collection days with 46.3% (801/1731) of landings occurring in the presence of an emanator and 53.7% (930/1,731) during control periods (Fig 2B). In a GLMM including distance as a continuous factor and controlling for environmental variables, location had no significant effect on *Ae. albopictus* HLC (P = 0.940), Table 2A. Also, there was no significant interaction between presence of treatment and distance as a continuous variable on *Ae. albopictus* HLC (P = 0.777), Table 2A. Factoring distance and controlling for environmental factors, there was a significant interaction between treatment and distance only at 5 m (P = 0.014), Table 2B. For the interaction of treatment and distances of 6m and 7m, there was a marginally significant effect (P<0.06), Table 2B.

After factoring distance and controlling for environmental factors, there was a significant interaction between treatment and distance at 5 m (P = 0.014), Table 2B. For distances of 6m and 7m, the effect of emanators was marginally significant (P<0.06), suggesting a potential effect, Table 2B. Of the environmental factors tested, there was a 1% increase in landing counts per percentage increase in humidity (P = 0.045), Table 2B.

## Study 3. Testing the effect of metofluthrin emanators on entry behavior

We recorded 436 landing counts over the course of four data collection days with 23.8% of all landings occurring in the presence of an emanator and 76.1% of all landings occurring during control periods. The mean landing count of the control was at least 1.7 times the mean landing count of the emanator (Fig 2C). Controlling for humidity, temperature, and wind speed, a GLMM showed metofluthrin emanators significantly reduced *Ae. albopictus* entry to the tent and subsequent landing with a 74.6% reduction in landing counts recorded in the presence of an emanator (P<0.001; Fig 2C), Table 3. Humidity, temperature, and wind speed were included in the model to reduce variance between days. Of the environmental factors tested, the model presented that there was a 7.5% increase in landing counts per one unit increase of wind speed in miles per hour (P = 0.0046), Table 3.

## Study 4. Testing emanator longevity and range of activity indoors

We recorded 2,885 HLC over the course of seven collection days with 42.8% of all landings occurring in the presence of an emanator and 57.2% occurring during control periods. Average landing counts during control periods was higher than in the presence of an emanator at both 1 and 3 m (Fig 3A). After accounting for distance and number of mosquitoes, the time mosquitoes were exposed to an emanator was a significant predictor of HLCs, Table 4A. After 30 minutes of exposure to the emanator, each additional minute of exposure led to a 2% reduction in mosquito landings (IRR = -0.02; P = 0.013), Table 4A. HLCs were reduced at 60 minutes of exposure in comparison to 30 minutes of exposure (Fig 3B). A GLMM quantified that

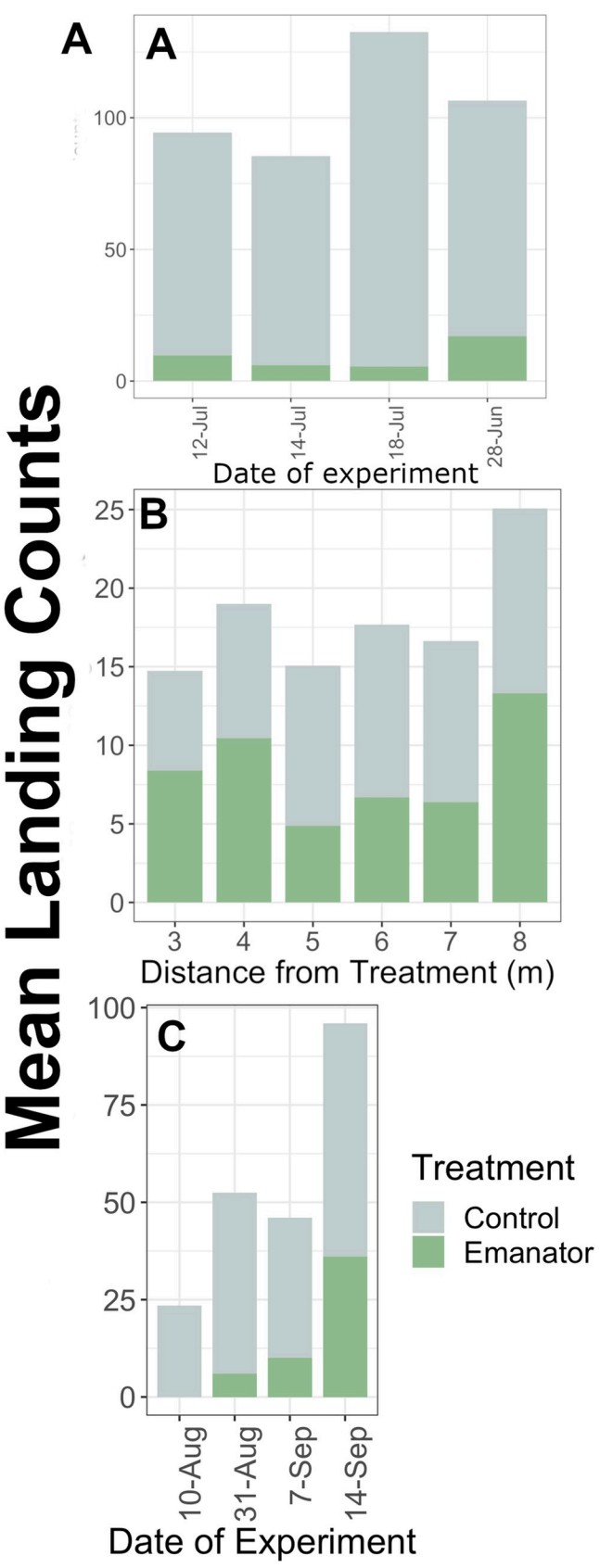

**Fig 2. Mean landing counts of each treatment by date and distance to the emanator in outdoors trials.** (A) In Study 1, the effect of the emanator on landing counts when placed next to a human volunteer was assessed (B) In Study 2, the effect of the emanator when placed at variable distances from a human volunteer was measured. (C) In Study 3, the effect of the emanator when placed inside a tent with a human volunteer sitting next to it was tested.

the emanator reduced landing counts by 61.1% in mosquitoes that had been exposed for at least 60 minutes (P<0.001), Table 4B. The interaction variable of treatment and distance showed no significance (P = 0.4114; Fig 3C), Table 4A.

Accounting for day and mosquito group, a CPHRM found that the probability of mortality among *Ae. albopictus* exposed to the emanator was 1.72 times higher than for *Ae. albopictus* exposed to the control in the control group (P = 0.0120), Table 5. The impact of distance from an emanator on the probability of mosquito mortality was assessed on a subset of the original data that included only mosquitoes exposed to the emanator. Accounting for mosquito group and collection day, distance from the emanator (1 to 3 m) did not impact mosquito mortality (P = 0.3672), Table 5.

The 95% confidence interval of the prediction of the reduction in landings as emanator age increased led to an estimate maximum residual life of 2.5 weeks, with average reduction in HLCs of 1.5 weeks (Fig 4). Assessment of model fit confirmed that the correct smoothness parameter was selected with a statistically significant p-value (Landings~Age, F = 4.766, P = 0.0262).

No KDC were recorded at 30 minutes of emanator exposure. Thus, the first model assessing the treatment's impact on mosquito knock downs included mosquitoes exposed to the treatments after 60 minutes. The emanator significantly knocked down *Ae. albopictus* mosquitoes (P<0.001), Table 5. This result was consistent with data collection as no knock downs occurred without the emanator. Distance from the emanator did not significantly impact the likelihood of an *Ae. albopictus* mosquito being knocked down (P = 0.4233).

Accounting for day and mosquito group, a CPHRM found that the probability of mortality among *Ae. albopictus* exposed to the emanator was 1.72 times higher than for *Ae. albopictus* exposed to the control in the control group (P = 0.0120), Table 6. The impact of distance from an emanator on the probability of mortality was assessed on a subset of the original data that included only mosquitoes exposed to the emanator. Accounting for mosquito group and collection day, distance from the emanator (1 to 3 m) did not impact mosquito mortality (P = 0.3672), Table 6.

The 95% confidence interval of the prediction of the reduction in landings as emanator age increased led to an estimate maximum residual life of 2.5 weeks, with average reduction of 1.5 weeks (Fig 4A). Assessment of model fit confirmed that the correct smoothness parameter was selected with a statistically significant p-value (Landings~Age, F = 4.766, P = 0.0262). Knock-down was similarly affected, with 95% confidence intervals predicting a maximum residual effect up to 3.8 weeks and an average effect of 3.1 weeks (Fig 4B). Model fit was confirmed by a statistically significant p-value (Knock Downs ~ Age, F = 4.897, P = 0.00566).

**Table 1. Parameter estimates of *Aedes albopictus* HLC post-exposure to an emanator placed next to human outdoors.**

| Parameter | Value | Standard Error | Z-value | P-value | 1-IRR |
|---|---|---|---|---|---|
| Intercept | 4.5404 | 0.0809 | 56.1 | **<0.001** | |
| Treatment (Emanator) | -2.2555 | 0.251 | -8.99 | **<0.001** | 0.895 |

In parenthesis is listed the variable used as baseline for effect estimation. For treatments showing a significant reduction in HLC, we estimated the efficacy, as 1—Incidence Rate Ratios (IRR). Bolded p-values are less than 0.05 and statistically significant.

**Table 2. IRR estimated from GLMM of landing counts upon exposure to emanators at different distances outdoors.**

| Model | Parameter | Value | Standard Error | Z-value | P-value |
|---|---|---|---|---|---|
| A. Distance as a continuous variable | Intercept | -1.74125 | 3.2169 | -0.54 | 0.588 |
| | Treatment (Control) | 0.02895 | 0.38717 | 0.07 | 0.940 |
| | Distance | 0.05326 | 0.04745 | 1.12 | 0.262 |
| | Humidity | 0.03819 | 0.01908 | 2.00 | **0.045** |
| | Temperature | 0.00716 | 0.07698 | 0.09 | 0.926 |
| | Wind Speed | 0.07679 | 0.06897 | 1.11 | 0.266 |
| | Treatment*Distance | 0.1754 | 0.06188 | 0.28 | 0.777 |
| B. Distance as a factor | Intercept | -0.6831 | 3.0854 | -0.22 | 0.825 |
| | Treatment (Control) | -0.3552 | 0.2913 | -1.22 | 0.223 |
| | Temperature | -0.0168 | 0.0735 | -0.23 | 0.819 |
| | Humidity | 0.0373 | 0.0186 | 2.00 | **0.045** |
| | Wind Speed | 0.0679 | 0.0677 | 1.00 | 0.316 |
| | Distance (4m) | 0.1421 | 2.899 | 2.78 | **0.0055** |
| | Distance (5 m) | -0.313 | 0.134 | -2.33 | **0.0199** |
| | Distance (6 m) | -0.1885 | 0.2601 | -0.72 | 0.469 |
| | Distance (7 m) | -0.2446 | 0.2637 | -0.93 | 0.354 |
| | Distance (8 m) | 0.4664 | 0.2254 | 2.07 | **0.039** |
| | Treatment*Distance (4 m) | 0.2803 | 0.3506 | 0.80 | 0.424 |
| | Treatment*Distance (5 m) | 0.9479 | 0.3506 | 0.80 | **0.014** |
| | Treatment*Distance (6 m) | 0.6892 | 0.3634 | 1.90 | 0.058 |
| | Treatment*Distance (7 m) | 0.7085 | 0.3693 | 1.92 | 0.055 |
| | Treatment*Distance (8 m) | 0.0663 | 0.3373 | 0.20 | 0.844 |

In parenthesis is listed the variable used as baseline for effect estimation. Bolded p-values are less than 0.05 and statistically significant.

## Discussion

Extending findings observed for *Ae. aegypti* [33] we show that a 10% metofluthrin passive emanator has high potential to reduce human-*Ae. albopictus* contacts both indoors and outdoors. Our study expands upon laboratory evaluations of metofluthrin passive emanators against *Ae. albopictus* by conducting studies under more realistic indoor and outdoor settings with a wild mosquito population that showed susceptibility to all insecticide classes.

In most settings, and particularly in the US, *Ae. albopictus* is a peridomestic vector that aggressively bites humans and other vertebrate hosts outdoors. In Atlanta and nearby locations, *Ae. albopictus* is the dominant mosquito species in residential areas [39] very often reaching high densities. As observed in our study, it is very common for observers to

**Table 3. IRR estimated from a GLMM with parameters affecting mosquito premise entry and landing behavior.**

| Parameter | Value | Standard Error | Z-value | P-value | 1-IRR |
|---|---|---|---|---|---|
| Intercept | -205.323 | 74.577 | -2.75 | **<0.001** | |
| Treatment (Emanator) | -1.369 | 0.323 | -4.24 | **<0.001** | 0.746 |
| Temperature | 8.053 | 2.899 | 2.78 | **0.0055** | |
| Humidity | -0.313 | 0.134 | -2.33 | **0.0199** | |
| Wind Speed | 2.02 | 0.713 | 2.83 | **0.0046** | |

In parenthesis is listed the variable used as baseline for effect estimation. For treatments showing a significant reduction in HLC, we estimated the efficacy, as 1—Incidence Rate Ratios (IRR). Bolded p-values are less than 0.05 and statistically significant.

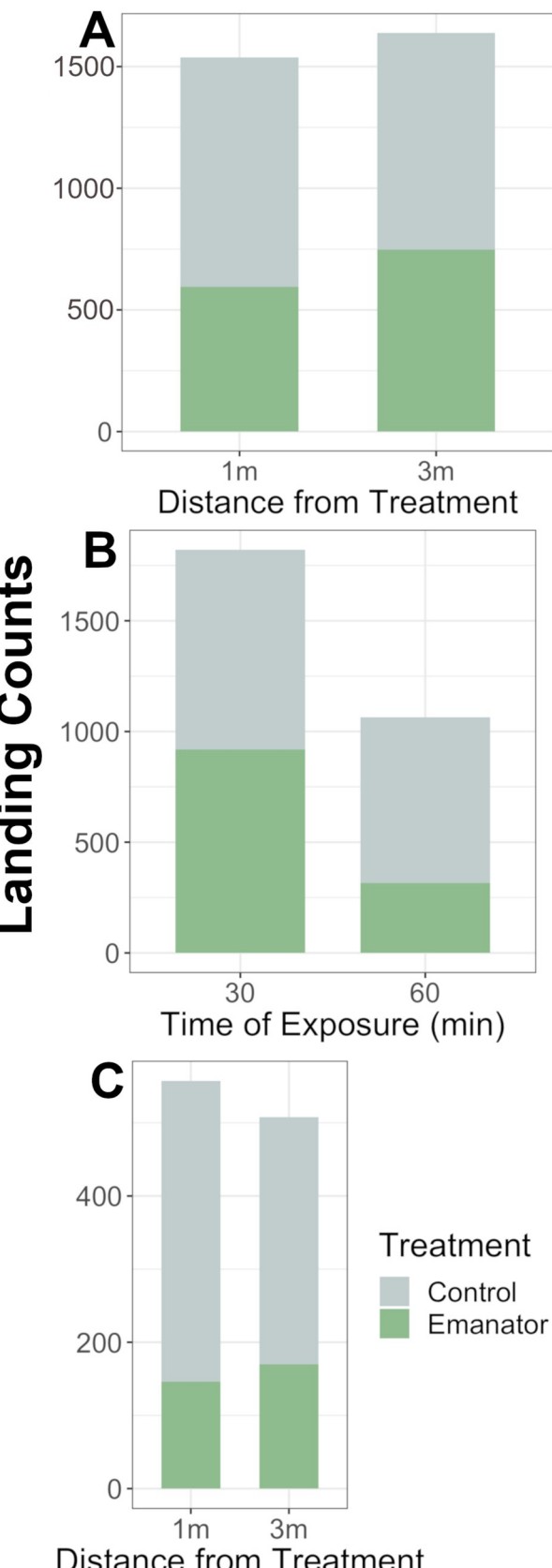

**Fig 3. Mean landing counts by time of exposure and distance from treatment indoors (Study 3).** (A) Emanator range of impact on *Ae. albopictus*. (B) Effect of emanator by time of exposure to emanator. (C) Effect of emanator by distance from treatment after 60 minutes of exposure.

experience more than 100 landings in 5 minutes of outdoor exposure. Application of barrier insecticide sprays in low vegetation within yards alone or in combination with larval control has been shown to be effective in reducing *Ae. albopictus* density [40]. Unfortunately, the high costs and large environmental impacts (particularly on non-target species) of widespread barrier spraying in residential areas may limit its scalability in highly urbanized regions of the US and Europe. In areas prone to *Ae. albopictus* invasion or at first detection, such approaches may be justified [41]. However, areas like Atlanta, where *Ae. albopictus* is fully established, such barrier insecticide sprays may not be feasible beyond privately contracted pest control services.

Minimizing human-mosquito contacts using passive metofluthrin emanators may provide an environmentally friendly personal protection alternative to reduce the nuisance and risk of *Ae. albopictus*. Our study shows over 90% reduction in landings when a 10% metofluthrin emanator is placed outdoors next to a human volunteer. Furthermore, while marginal reductions were observed at increasing distances of the emanator (up to 7 meters), the efficacy of the product may be increased by having a product with higher % of a.i. (the prototype we used was a 10% w/w formulation but alternatives with higher a.i. might be engineered). Further, commercial metofluthrin products exist (e.g., Off! Clip-on, an active emanator, SJ Johnsons, Racine, WI) and their efficacy against *Ae. albopictus* was slightly lower when worn by human volunteers outdoors (70% reduction, versus 90% reduction in our study) [42]. In situation where people may be sitting outdoors (patios, outside tables in restaurants, porches) passive metofluthrin emanators may provide important protection with minimal insecticide use. The efficacy and range of possible applications of metofluthrin passive emanators remains to be studied.

Product longevity is key for the proper adoption of an intervention. For *Ae. aegypti*, the same product we evaluated provided protection for up to 21 days in experimental rooms [32]. We found that for *Ae. albopictus* metofluthrin emanators significantly reduced HLCs indoors up to 2.5 weeks from deployment. A similar study indicated that continuous use of a spatial

**Table 4. IRR estimates from a GLMM Model of mosquito landing counts upon exposure to metofluthrin emanators at different distances indoors.**

| Model | Parameter | Value | Standard Error | Z-value | *P*-value | *1-IRR* |
|---|---|---|---|---|---|---|
| A. Determining impact of time on landing rates. | Intercept | -5.90479 | 0.74744 | -7.90 | <**0.001** | **0.02** |
| | Treatment (Emanator) | 0.77661 | 0.40381 | 1.92 | 0.054 | |
| | Time | -0.00523 | 0.00594 | -0.88 | 0.378 | |
| | Distance (3 m) | 1.91 | 0.12917 | 1.48 | 0.140 | |
| | Treatment*Time | -0.02157 | 0.00872 | -2.47 | **0.013** | |
| B. Assessing impact of distance on effect of treatment after 60 minutes of treatment exposure. | Intercept | -6.0017 | 0.6678 | -8.99 | <**0.001** | **0.611** |
| | Treatment (Emanator) | -0.9452 | 0.2809 | -3.36 | <**0.001** | |
| | Distance (3 m) | 0.0127 | 0.2575 | 0.05 | 0.9607 | |
| | Treatment*Distance | 0.3109 | 0.3785 | 0.82 | 0.4114 | |

In parenthesis is listed the variable used as baseline for effect estimation. For treatments showing a significant reduction in HLC, we estimated the efficacy, as 1—Incidence Rate Ratios (IRR). Bolded p-values are less than 0.05 and statistically significant.

**Table 5. Estimates from a CPHM of parameters affecting mosquito mortality upon exposure to emanators indoors.**

| Parameter | Value | Standard Error | Z-value | P-value |
|---|---|---|---|---|
| Treatment (Control) | -0.5697 | 0.2268 | -2.51 | **0.0120** |
| Distance | -0.0518 | 0.05741 | -0.90 | 0.3672 |
| Treatment (Control)*Distance | 0.2047 | 0.09815 | 2.09 | **0.0370** |

Day B is 11/10/17, Day C is 11/17/17, Day D is 11/25/17, Day E is 12/1/17, Day F is 12/9/17. In parenthesis is listed the variable used as baseline for effect estimation. Bolded p-values are less than 0.05 and statistically significant.

metofluthrin repellent device is efficacious for about 2–3 weeks outdoors [43]. As an important fraction of the a.i. remains in the polyacrylamide mesh [29], novel engineering approaches may lead to longer duration of the emanator efficacy if they can minimize a.i. retention by the mesh. Furthermore, compared to metofluthrin coils, passive emanators have significantly

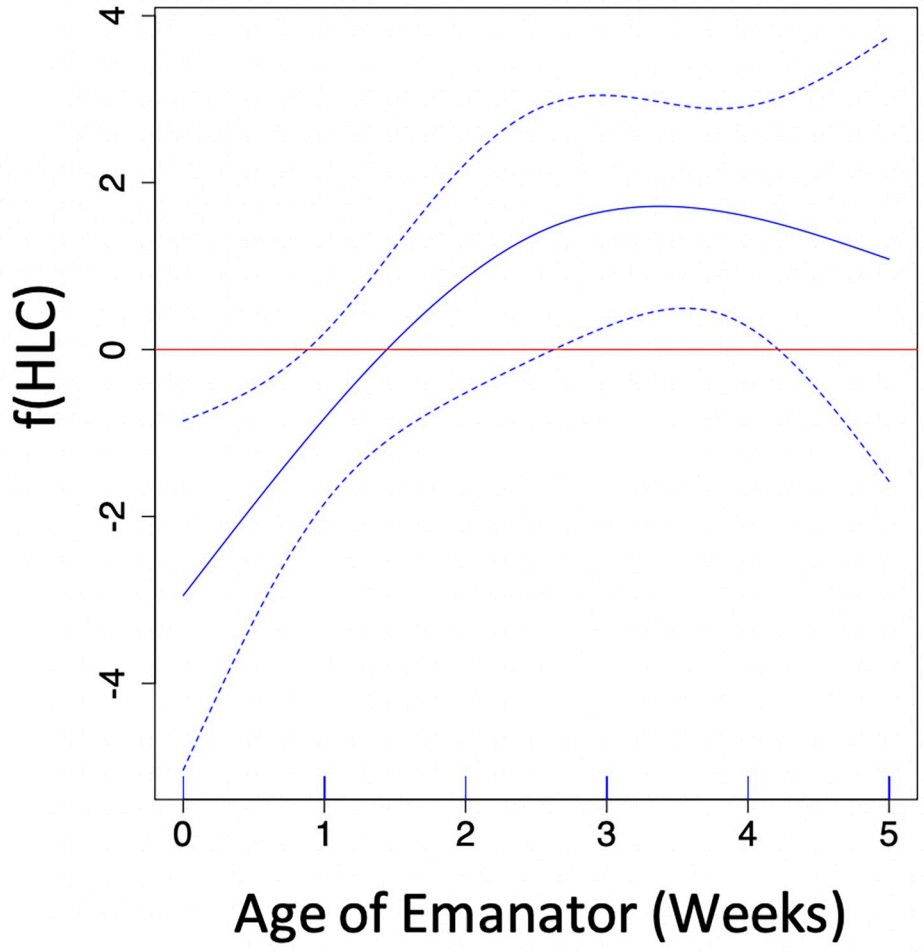

**Fig 4. Modeling the impact of emanator age on HLC.** General additive model results from the trial testing range of effect of metofluthrin emanators on *Ae. albopictus* landing counts (*f(HLC)*) indoors compared to a trial without emanator. The zero line shows the no-effect threshold and dotted lines the 95% confidence interval of *f(HLC)*. Anything above the line shows significant increase in HLCs compared to control, anything below the line shows the opposite.

**Table 6. Parameters from Cox Proportional Hazards Regression Model affecting *Aedes albopictus* mortality post-exposure to emanators indoors.**

| Parameter | Value | Standard Error | Z-value | *P*-value |
|---|---|---|---|---|
| Treatment (Control) | -0.5697 | 0.2268 | -2.51 | **0.0120** |
| Distance | -0.0518 | 0.05741 | -0.90 | 0.3672 |
| Treatment (Control)*Distance | 0.2047 | 0.09815 | 2.09 | **0.0370** |

The variable used as baseline for effect estimation is listed in parenthesis. Bolded p-values are less than 0.05 and statistically significant.

longer duration and the added benefit of not requiring burning or other source of heat. In Merida, Mexico, household surveys showed that people view very positively the use of metofluthrin emanators as alternative to coils and reported measurable reductions in mosquito bites after their use [33].

Previous research determined that ten minutes of exposure indoors reduced *Ae. aegypti* landing counts [32, 26]. Research indicated that the devices' effectiveness increases the longer it is implemented likely due to the slow dispersal of the molecule into an environment [26]. The reduction in landings we observed at one hour coincided with an increased number of mosquitoes exhibiting intoxication signs such as paralysis. Such effect was larger after 2 hours of exposure. Therefore, implementation guidelines should recommend the device be hung in an indoor ventilated location for at least one hour to ensure high protective efficacy to those entering or residing in it.

There were several limitations in our study. In outdoor experiments, while we controlled for environmental conditions in our analysis (temperature, humidity, wind) we could not sustain similar environmental conditions, which may explain why some days had different reductions in landings than others. Repeating the experiment for more days would have shown stronger effects, particularly when evaluating the influence of emanators at increasing distances outdoors. The marginal associations found at 6–7 m may be influence of prevailing wind or other factors, and the experiment should be repeated to ensure the actual range of efficacy outdoors. Unfortunately, limited number of emanators prevented us from repeating that study. The impact of UV sunlight on metofluthrin efficacy was not considered in our study. Recent research has tested the impact of environmental conditions on the protective ability of metofluthrin through building modular wind tunnels outfitted with UV light bulbs [44]. The results from such study indicated that 15 to 20 minutes of UV light exposure removed the spatial repellent to a level that lessened the effect on mosquito biting and flying behavior [44].

Also, the time the emanator was evaluated outdoors (5 min) may have not been sufficient. A study evaluating metofluthrin active emanator Off! Clip-on evaluated efficacy over a 3h period [43]. Extending beyond 5 minutes may have provided a temporal signal of the effect of metofluthrin on mosquito attraction, biting and repellency. Metofluthrin not only repels mosquitoes but also works as a "confusant" with additional lethal impacts [32]. In our trial the behavioral impacts of metofluthrin (which were landing counts and reduced entry to tents) were evident after 60 minutes, but future experiments should determine if mosquitoes repelled by the device are later knocked down.

In our current study *Ae. albopictus* exposed to a 10% metofluthrin emanator indoors reduced landings by 70% regardless of the distance to the emanator within a regular bedroom (3m), showing potent efficacy as reported in other studies. A recent entomological randomized field trial of the same emanator prototype in Merida, Mexico, showed that when placed at a rate of one per bedroom, emanators reduced indoor *Ae. aegypti* abundance by 70% [33]. Our current experimental findings coupled with the field trial results for *Ae. aegypti* [33] provide enough evidence to support the design of further trials for *Ae. albopictus*. The rapid invasion

of *Ae. albopictus* in Europe, with subsequent recent occurrence of arbovirus outbreaks [17] requires vector control methods that can be effective and scalable. If efficacious, metofluthrin emanators could be deployed in affected areas (for instance, during outbreaks) for rapid reduction of human-mosquito contacts and disease. Provided mosquitoes are susceptible to the a.i., passive emanators can fill an important niche in urban control of *Aedes*-transmitted arbovirus.

## Supporting information

**S1 File.**
(DOCX)

## Acknowledgments

We would like to thank to graduate and undergraduate students from the Department of Environmental sciences, Emory University, for their collaboration in data collection.

## Author Contributions

**Conceptualization:** Olivia Zarella, Uche Ekwomadu, Pablo Manrique-Saide, Gregor Devine, Gonzalo M. Vazquez-Prokopec.

**Data curation:** Olivia Zarella.

**Formal analysis:** Olivia Zarella, Uche Ekwomadu, Gabriela González-Olvera, Gonzalo M. Vazquez-Prokopec.

**Funding acquisition:** Gregor Devine.

**Investigation:** Gabriela González-Olvera, Pablo Manrique-Saide, Gregor Devine, Gonzalo M. Vazquez-Prokopec.

**Methodology:** Olivia Zarella, Uche Ekwomadu, Yamila Romer, Oscar D. Kirstein, Azael Che-Mendoza.

**Resources:** Yamila Romer, Oscar D. Kirstein, Azael Che-Mendoza, Gabriela González-Olvera.

**Writing – original draft:** Olivia Zarella, Gonzalo M. Vazquez-Prokopec.

**Writing – review & editing:** Olivia Zarella, Uche Ekwomadu, Azael Che-Mendoza, Gabriela González-Olvera, Pablo Manrique-Saide, Gregor Devine, Gonzalo M. Vazquez-Prokopec.

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
