## [Decision Letter · Decision Letter 0]

29 Sep 2021

PONE-D-21-25159Experimental evaluation of a metofluthrin passive emanator against Aedes albopictusPLOS ONE

Dear Dr. Gonzalo M Vazquez-Prokopec,

Thank you for submitting your manuscript to PLOS ONE. After careful consideration, we feel that it has merit but does not fully meet PLOS ONE’s publication criteria as it currently stands. Therefore, we invite you to submit a revised version of the manuscript that addresses the points raised during the review process.

ACADEMIC EDITOR: This study includes human in the experiment, so, ethical approve should be included in the manuscript.

We look forward to receiving your revised manuscript.

Kind regards,

Shawky M. Aboelhadid, PhD

Academic Editor

PLOS ONE

3. Please note that supplementary tables should be uploaded as separate "supporting information" files"

Reviewers' comments:

Reviewer's Responses to Questions

**Comments to the Author**

1. Is the manuscript technically sound, and do the data support the conclusions?

Reviewer #1: Yes

Reviewer #2: Yes

2. Has the statistical analysis been performed appropriately and rigorously? 

Reviewer #1: Yes

Reviewer #2: Yes

3. Have the authors made all data underlying the findings in their manuscript fully available?

Reviewer #1: Yes

Reviewer #2: Yes

4. Is the manuscript presented in an intelligible fashion and written in standard English?

Reviewer #1: Yes

Reviewer #2: Yes

5. Review Comments to the Author

Reviewer #1: It is a comprehensive study on metofluthion against Aedes mosquitoes even if there are so many studies and literatures. This report provided useful information. However, I did not find the provided number about using human subjects for the landing rate counts and the approved animal care number for the blood feeding for the lab colony.

Bibbs CS did several years study about the same compound and published several important articles should be checked and cited. J. Med. Entomol. 53:480-487,2016, 57:17-24, 2020, and Environ Entomol. 49:435-443.

Khater EIM et al. 2021. Insecticide efficacy of spatial repellent compound metofluthrin against susceptible and resistant strains of Aedes aegypti. Journal of the Florida Mosquito Control Association 68:86-91. DOI.org/10.32473/JFMCA.v68i1.129104

Reviewer #2: The paper is well written. It provided background information on the efficacy of metofluthrin-based devices on mosquito and the knowledge gap regarding metofluthrin-based emanators on Aedes albopictus. The findings of the study provide valuable information for personal protection against Ae. albopictus using the emanators. The limitations of the study provide guidance in interpreting and using the information generated as well as pointers for further studies.

A few repetitions were noted in the manuscript and these have been indicated for the authors to address. Other minor comments/corrections have been indicated. The x-axis caption of Fig. 2A is not legible.

Of concern is the absence of ethical clearance for this study since it involved humans. In Human Landing Catch (HLC) experiments, human volunteers are used as baits and hence there is the need for ethical clearance for such studies. The authors should indicate why ethical clearance was not sought if that is the case.

6. PLOS authors have the option to publish the peer review history of their article (what does this mean?). If published, this will include your full peer review and any attached files.

Reviewer #1: **Yes: **Rui-De Xue

Reviewer #2: No

---

## [Author Response · Author response to Decision Letter 0]

28 Mar 2022

Reviewers' comments:

Reviewer #1: It is a comprehensive study on metofluthion against Aedes mosquitoes even if there are so many studies and literatures. This report provided useful information. However, I did not find the provided number about using human subjects for the landing rate counts and the approved animal care number for the blood feeding for the lab colony.

Bibbs CS did several years study about the same compound and published several important articles should be checked and cited. J. Med. Entomol. 53:480-487,2016, 57:17-24, 2020, and Environ Entomol. 49:435-443.

Khater EIM et al. 2021. Insecticide efficacy of spatial repellent compound metofluthrin against susceptible and resistant strains of Aedes aegypti. Journal of the Florida Mosquito Control Association 68:86-91. DOI.org/10.32473/JFMCA.v68i1.129104

R. Thank you for the references. Additionally, we provide Emory’s IRB code, now reviewed and approved by the IRB. Per Emory University procedures, we are not required to obtain approval to use heparinized rabbit blood for feeding laboratory-reared mosquitoes. Our lab has all approvals from Environmental Health Office required for mosquito rearing. No IACUC is required in that case. 

Reviewer #2: The paper is well written. It provided background information on the efficacy of metofluthrin-based devices on mosquito and the knowledge gap regarding metofluthrin-based emanators on Aedes albopictus. The findings of the study provide valuable information for personal protection against Ae. albopictus using the emanators. The limitations of the study provide guidance in interpreting and using the information generated as well as pointers for further studies.

A few repetitions were noted in the manuscript and these have been indicated for the authors to address. Other minor comments/corrections have been indicated. The x-axis caption of Fig. 2A is not legible.

R. We have taken them all into consideration. Thank you.

Of concern is the absence of ethical clearance for this study since it involved humans. In Human Landing Catch (HLC) experiments, human volunteers are used as baits and hence there is the need for ethical clearance for such studies. The authors should indicate why ethical clearance was not sought if that is the case.

R. We have obtained ethical clearance and now we list it in the method section.

---

## [Decision Letter · Decision Letter 1]

6 Apr 2022

Experimental evaluation of a metofluthrin passive emanator against Aedes albopictus

PONE-D-21-25159R1

Dear Dr. Gonzalo,

We’re pleased to inform you that your manuscript has been judged scientifically suitable for publication and will be formally accepted for publication once it meets all outstanding technical requirements.

Kind regards,

Shawky M Aboelhadid, PhD

Academic Editor

PLOS ONE

Additional Editor Comments (optional):

Reviewers' comments:

Reviewer's Responses to Questions

**Comments to the Author**

1. If the authors have adequately addressed your comments raised in a previous round of review and you feel that this manuscript is now acceptable for publication, you may indicate that here to bypass the “Comments to the Author” section, enter your conflict of interest statement in the “Confidential to Editor” section, and submit your "Accept" recommendation.

Reviewer #1: All comments have been addressed

2. Is the manuscript technically sound, and do the data support the conclusions?

Reviewer #1: Yes

3. Has the statistical analysis been performed appropriately and rigorously? 

Reviewer #1: Yes

4. Have the authors made all data underlying the findings in their manuscript fully available?

Reviewer #1: Yes

5. Is the manuscript presented in an intelligible fashion and written in standard English?

Reviewer #1: Yes

6. Review Comments to the Author

Reviewer #1: I am ok with the authors' responses and answers to my comments and suggestion.

7. PLOS authors have the option to publish the peer review history of their article (what does this mean?). If published, this will include your full peer review and any attached files.

Reviewer #1: No

---

## [Editor Report · Acceptance letter]

29 Apr 2022

PONE-D-21-25159R1 

Experimental evaluation of a metofluthrin passive emanator against *Aedes albopictus*

Dear Dr. Vazquez-Prokopec:

I'm pleased to inform you that your manuscript has been deemed suitable for publication in PLOS ONE. Congratulations! Your manuscript is now with our production department. 

Kind regards, 

on behalf of

Professor Shawky M Aboelhadid 

Academic Editor

PLOS ONE